# Scaling Synthetic Brain Data Generation

Mike Doan
*Department of Computer Science*
*Georgia State University*
*TReNDS Center*
Atlanta, GA, USA
orcid.org/0009-0008-0703-8585

Sergey Plis
*Department of Computer Science*
*Georgia State University*
*TReNDS Center*
Atlanta, GA, USA
orcid.org/0000-0003-0040-0365

*Abstract*—The limited availability of diverse, high-quality datasets is a significant challenge in applying deep learning to neuroimaging research. Although synthetic data generation can potentially address this issue, on-the-fly generation is computationally demanding, while training on pre-generated data is inflexible and may incur high storage costs. We introduce Wirehead, a scalable in-memory data pipeline that significantly improves the performance of on-the-fly synthetic data generation for deep learning in neuroimaging. Wirehead's architecture decouples data generation from training by running multiple generators in independent parallel processes, facilitating near-linear performance gains proportional to the number of generators used. It efficiently handles terabytes of data using MongoDB, greatly minimizing prohibitive storage costs. The robust, modular design enables flexible pipeline configurations and fault-tolerant operation. We evaluated Wirehead with SynthSeg, a synthetic brain segmentation data generation tool that requires 7 days to train a model. When deployed in parallel, Wirehead achieved a near-linear 15.7x increase in throughput with 16 generators. With 20 generators, we can train a model in 9 hours instead of 7 days. This demonstrates Wirehead's ability to greatly accelerate experimentation cycles. While Wirehead represents a substantial step forward, it also reveals opportunities for future research in optimizing generation-training balance and resource allocation. Its ability to facilitate distributed deep learning has significant implications for enabling more ambitious neuroimaging research.

*Index Terms*—deep learning, synthetic data generation, distributed computing, neuroimaging, magnetic resonance imaging

## I. Introduction

Deep learning has revolutionized many domains, including neuroimaging, computer vision, and natural language processing. However, training these models efficiently often faces a critical bottleneck: the delay in feeding data to the GPU. This issue is particularly pronounced in data-parallel training across multiple GPUs, when preprocessing training samples, and when generating training data in real-time. In the latter case, data generation can take orders of magnitude longer than the actual GPU training step.

For example, in neuroimaging, generating synthetic brain MRI scans for training deep learning models can be extremely time- and resource-intensive. Billot et al. [1], [2] reported that generating and training on $300,000$ synthetic scans using their SynthSeg, a synthetic MRI data generation tool, on a single

Manuscript submitted Oct 22, 2024. This work was supported by NIH grants R01MH129047 and 2R01EB006841, and NSF grant 2112455.

computer took 7 days. Storing these scans would require an astonishing 91 TB at uint8 precision or 366 TB at float32, making on-disk storage practically infeasible. The same core generation process is used in a number of other papers with slight modification on how data is generated to fit the task [3]–[6]. Training models for a week does not leave much time for experimentation, hyper-parameter search, and model architecture innovation. This severely limits iteration speed and flexibility in model development. Remarkably, the result of this limitation is that all of these models use practically the same architecture for their fully convolutional network workhorse: the U-Net [7], [8].

This 7 day experimentation cycle has slowed down architectural innovation, although not the variety of applications, but we attribute it to the great need and success of this particular generator. Yet, the generator is relatively simple methodologically as its core computational component is a Gaussian mixture model (GMM). However, the field of developing brain simulations is blooming with more complex, potentially more advanced, and highly computationally sophisticated methods such as diffusion models [9]–[11], generative adversarial networks (GANs) [12], [13], and many more [14]. Potentially, these rising generative models for synthetic brain images can be more flexible and provide ways to train models for biomedical applications where data is although specific, such as stroke lesions, but relatively rare. However, these models would be substantially slower to run and much more difficult to train on their data if the approach to training on a single GPU sequentially of Billot et al. [1], [2] is maintained.

To address these challenges, we introduce Wirehead, a novel in-memory data pipeline that enables model-agnostic scalable, real-time synthetic data generation for deep learning. Wirehead's key innovations include a decoupled architecture for independent scaling of data generation and training, an efficient MongoDB-based [15] data management system, and a robust, fault-tolerant design.

By enabling distributed synthetic data generation and reducing storage costs, Wirehead significantly accelerates the deep learning experimentation cycle. While we demonstrate Wirehead's application to neuroimaging using SynthSeg, our approach is broadly applicable to any domain requiring large-scale on-the-fly generation of synthetic data for training.

In this paper, we first present Wirehead's design principles

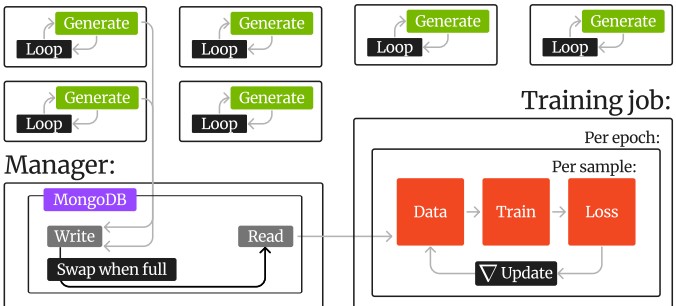

(a) Wirehead decoupled architecture for training and data generation

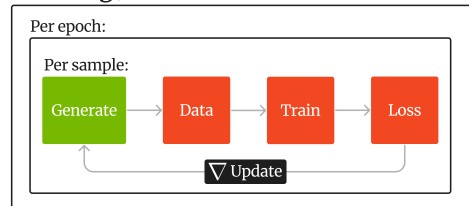

(b) Traditional coupled training and data generation

Fig. 1: Comparison of architectures for data generation and training: (a) Wirehead decoupled architecture, (b) Traditional coupled architecture. In (a), training is unaware of the rest of the process, both in the code and the interface. It continuously trains from what appears to be the same dataset.

and system architecture (Section II). We then detail our data management strategy (Section III) and the decoupling of data generation and model training (Section IV). Next, we report extensive benchmarks evaluating Wirehead's performance and scalability (Section V) and (Section VI). Finally, we discuss trade-offs, future directions, and conclude (Sections VII-X).

## II. DESIGN OVERVIEW

Wirehead's architecture splits the workflow into three primary components:

1) Generator: This component runs the synthesis, preprocessing, and scheduling in a continuous loop. Multiple generators can operate concurrently, each pushing data to a central database.
2) Manager: Connected to MongoDB, the manager oversees data flow and integrity. It partitions the database into read and write collections. When the write collection reaches capacity, the manager swaps it with the read collection, ensuring a constant supply of fresh data for training. The manager also handles data integrity checks and error recovery.
3) Dataset: This component interfaces with MongoDB, allowing data to be pulled for training just like standard PyTorch datasets. It includes safeguards against data corruption, ensuring robust operation even in the face of potential errors.

Central to Wirehead's design is the use of MongoDB. Chosen for its non-blocking I/O operations and ability to scale to terabytes of data, MongoDB enables efficient storage and movement of large datasets. This choice facilitates the asynchronous, decoupled nature of our system, where generators and training jobs can scale independently without creating bottlenecks.

For testing and deployment, Wirehead utilizes lightweight SLURM scripts, though the architecture supports local deployment and other alternatives. This flexibility allows for easy adaptation to various computational environments.

The decoupled, asynchronous design offers several advantages:

1) Independent Scaling: Generation and training processes can be scaled separately based on computational demands.
2) Continuous Operation: Neither component waits for the other, maximizing resource utilization.
3) Fault Tolerance: Issues in one component (e.g., a failing generator) don't immediately impact others.
4) Flexible Resource Allocation: Computational resources can be dynamically assigned to generation or training as needed.

By re-imagining the synthetic training pipeline as a distributed, decoupled system, Wirehead achieves high scalability and efficiency, particularly data-intensive for machine learning tasks.

## III. DATA MANAGEMENT

Wirehead employs an innovative data management system designed to handle large-scale synthetic data generation and training without the need for disk storage. At its core is a cache system divided into two halves: read and write. This design, coupled with an efficient swap mechanism, allows for continuous data generation and training while minimizing I/O bottlenecks.

### A. Cache Structure and Swap Timing

The cache consists of two main components:

1) Write Cache: Where newly generated samples are stored.
2) Read Cache: From which the training process pulls data.

The timing and size of data swaps between these caches are controlled by two factors:

1) Swap Cap: Determines the maximum number of samples in each cache half at any given time.
2) Sample Generation Rate: Dictates the frequency of swaps based on how quickly new samples are produced.

This dual-cache system ensures a constant supply of fresh data for training while allowing older samples to be cycled out efficiently.

## B. Efficient Swap Implementation

The swap operation is a critical engineering challenge in Wirehead. Naively copying hundreds of gigabytes of data between write and read caches would be prohibitively I/O intensive. Wirehead circumvents this issue with an ingenious solution:

Instead of moving data, the swap is executed by simply changing the collection identifier of the databases. This approach reduces the swap operation to O(1) complexity, regardless of the data volume involved.

The swap process consists of the following steps:

1) Renaming: The 'write' collection is renamed to a temporary 'temp' collection.
2) Data Integrity Verification: Checks are performed to ensure the write database contains the required number of samples and that all samples are complete.
3) Data Reindexing: Data chunks are reorganized into contiguous segments, optimizing subsequent read operations.
4) Final Renaming: The 'temp' collection is renamed to become the new 'read' collection, while the previous 'read' collection is dropped.

This process ensures that at any given time, the training process has access to a complete, integrity-verified set of data, while new samples continue to be generated and stored in the new 'write' collection.

## C. Benefits of the Approach

1) Minimal I/O Overhead: By avoiding large data transfers during swaps, Wirehead maintains high performance even with very large datasets.
2) Continuous Operation: The training process can continue uninterrupted during swaps, as it always has access to a complete 'read' collection.
3) Scalability: This approach scales efficiently with increasing data volumes, making it suitable for large-scale machine learning tasks.
4) Data Freshness: Regular swaps ensure that the training process always has access to recently generated data, which can be crucial in dynamic or evolving data generation scenarios.

Wirehead's data management system demonstrates that clever engineering can overcome significant scalability challenges in machine learning pipelines. By rethinking traditional data handling approaches, it achieves a level of efficiency that enables new scales of synthetic data generation and utilization in training processes.

## IV. DECOUPLED GENERATION AND TRAINING

A key innovation in Wirehead's architecture is the complete decoupling of data generation and model training processes. This design choice offers several significant advantages:

## A. Horizontal Scaling

The decoupled architecture allows for independent scaling of generation and training components. As computational demands change, resources can be added or removed from either process without affecting the other. This flexibility enables efficient resource utilization and easy adaptation to varying workloads.

## B. Continuous Operation

In Wirehead, no component waits for another, ensuring maximum utilization of computational resources:

1) Generation Pushes Out of Order: Generators continuously produce and push data to the database without waiting for any acknowledgment from the training process or other generators. This asynchronous operation allows for uninterrupted data creation, with insertions happening completely out of order as each generator works independently.
2) Isolated Read Collection: Training nodes pull data from an isolated read collection in a round-robin fashion. This separation ensures that the training process always has a consistent, integrity-verified dataset to work with, regardless of ongoing generation activities.

## C. Flexible Resource Allocation

The system allows for dynamic allocation of computational resources between generation and training. Depending on the specific needs of a project, more or less compute power can be assigned to either process. This flexibility is particularly valuable in scenarios where generation and training have different computational requirements or when requirements change over time.

## D. Fault Tolerance

Wirehead's architecture is designed with robustness in mind:

1) Generator Flexibility: The system can function effectively even if not all generators are active. This allows for maintenance or upgrades of individual generators without halting the entire process.
2) Manager Resilience: In the event of a manager failure, the system continues to operate. While new data may not be swapped in, the training process can continue with existing data, potentially leading to overfitting but avoiding complete system failure.
3) Training Robustness: The training process can pull samples independently of the manager's state, thanks to robust error handling. This includes handling the edge case where a read operation occurs mid-swap:
The '__getitem__' method in the dataset class is designed to handle failures that may occur when attempting to read data during a swap operation. If such a failure occurs, the method employs a retry mechanism, allowing the training process to continue smoothly once the swap is complete.

This decoupled, fault-tolerant design ensures that Wirehead can maintain operation even in the face of component failures

or maintenance events, providing a level of robustness crucial for long-running, large-scale machine learning tasks.

## V. RESULTS

To evaluate the performance and scalability of Wirehead, we conducted a series of benchmarks comparing different configurations. These tests offer insights into the system's efficiency and its ability to scale in distributed environments.

We tested four main configurations to assess Wirehead's performance. All experiments were conducted on a system with the following hardware configuration:

- GPU: 1x NVIDIA A100 (80GB)
- CPU: 16 cores
- RAM: 200 GB

The experiments were deployed and managed using the SLURM job scheduler. This setup allowed for consistent resource allocation and management during the benchmarking process.

1) Coupled training and generation
   - Generation and training live in the same process, and are mutually blocking.
   - Single node, sequential generation and training.
2) Wirehead on Local Machine
   - Decoupled generation and training, in which generation and training happen on different processes, and are non blocking.
   - Single node for both generation and training.
3) Wirehead on Distributed System
   - Basic distributed setup, in which generation and training happens on different nodes.
   - Multiple nodes for generation, single node for training.
4) Distributed Wirehead with Various Worker Configurations
   - Multiple configurations varying the number of generator nodes.
   - Tested with 2, 4, 8, and 16 generator nodes.

TABLE I: Samples Read Throughput by Configuration

| Experiment | Performance Metrics | | |
|---|---|---|---|
| Type | Samples/sec | Uncertainty | Speedup |
| Coupled | 0.21 | ± 0.01 | 1.00x |
| Local Wirehead | 0.74 | ± 0.18 | 3.45x |
| Distributed Wirehead | 0.75 | ± 0.06 | 3.51x |

TABLE II: Sample Generation Throughput by Configuration

| Experiment | Performance Metrics | | |
|---|---|---|---|
| Type | Samples/sec | Uncertainty | Speedup |
| Coupled | 0.21 | ± 0.01 | 1.00x |
| Local Wirehead | 0.21 | ± 0.00 | 1.00x |
| Distributed Wirehead | 0.25 | ± 0.00 | 1.18x |

TABLE III: Scaling Performance with Multiple Generators

| Configuration | Performance Metrics | | |
|---|---|---|---|
| Type | Workers | Samples/sec | Scale Factor |
| 1x Wirehead Generator | 1 | 0.25 | 1.00 |
| 2x Wirehead Generator | 2 | 0.50 | 1.99 |
| 4x Wirehead Generator | 4 | 0.98 | 3.91 |
| 8x Wirehead Generator | 8 | 1.95 | 7.75 |
| 16x Wirehead Generator | 16 | 3.94 | 15.70 |

## VI. ANALYSIS

Our analysis focuses on five key metrics:

### A. GPU Utilization

The effects of the manner of synthetic data generation on GPU utilization are demonstrated in Figure 2 and can be summarized as:

- Coupled: 34.76% average utilization
- Local Wirehead: 89.27% average utilization
- Distributed Wirehead: 87.48% average utilization

The GPU utilization metrics illustrate the hardware benefits of using Wirehead compared to the traditional coupled approach. In the coupled setup, the GPU is often idle while waiting for new data, resulting in low average utilization of just 33.70%. In contrast, Wirehead configurations show significant improvements: Local Wirehead achieves 93.21% average GPU utilization, while Distributed Wirehead maintains 88.84%. These results highlight Wirehead's ability to maximize GPU usage efficiency by decoupling data generation and training processes, leading to faster training times and improved performance.

The slightly higher GPU utilization in Local Wirehead can be attributed to the generation process partially utilizing the GPU and sharing resources with the training process. In Distributed Wirehead, the overhead of data transfer and separate generation machines result in slightly lower utilization. However, the benefits of distributed computation, notably the ability to linearly scale throughput by increasing generator count, outweigh these minor differences.

### B. Samples Read per Second

Wirehead's architecture allowed for an approximate 3.5x increase in training throughput compared to the coupled approach. The slightly higher throughput in Distributed Wirehead can be attributed to the absence of local data management overhead. However, this difference is minimal, and both Wirehead configurations provide significant improvements over the coupled setup.

The higher variance in Local Wirehead (0.74 ± 0.18) compared to Distributed Wirehead (0.75 ± 0.06) is likely due to resource contention between data generation and training processes on the same machine.

The overall sample reading speed is summarized in Table I.

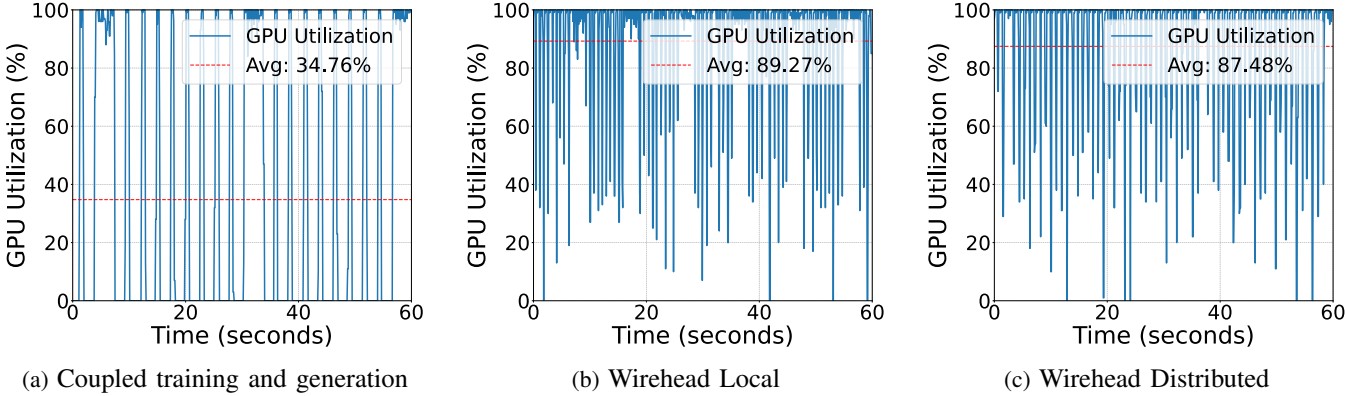

(a) Coupled training and generation     (b) Wirehead Local     (c) Wirehead Distributed

Fig. 2: GPU utilization during training for different configurations: (a) Coupled training and generation, (b) Wirehead Distributed, and (c) Wirehead Local. Consistent GPU utilization allows a model to train faster by minimizing the idle time.

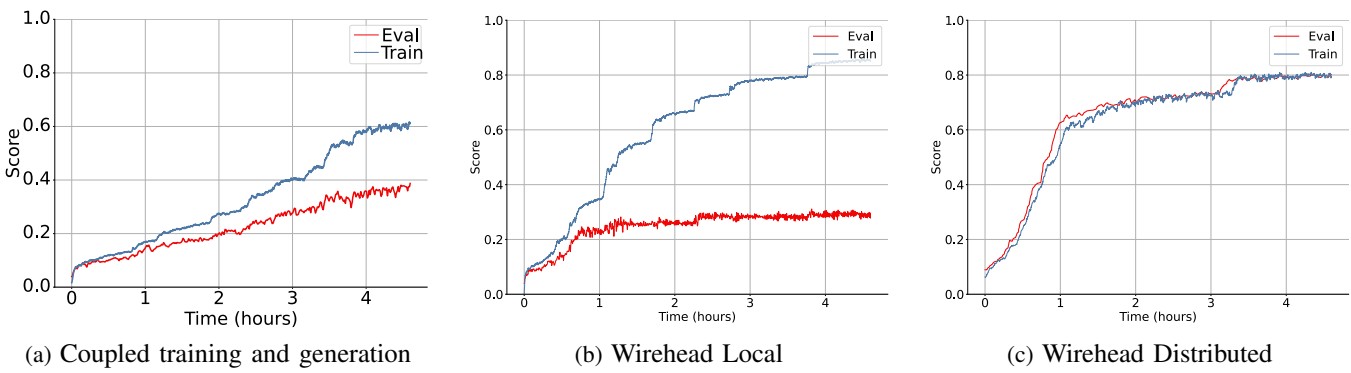

(a) Coupled training and generation     (b) Wirehead Local     (c) Wirehead Distributed

Fig. 3: Dice score for training and evaluation for different configurations: (a) Coupled training and generation, (b) Wirehead local, and (c) Wirehead distributed. The distributed version is using 20 generators simultaneously.

### C. Samples Generated per Second

Wirehead achieves a notable 19% increase in generation throughput when running in a distributed setting (0.25 samples/sec) compared to both the coupled and local Wirehead approach (0.21 samples/sec), while maintaining the same number of generators.

Local Wirehead maintains the same generation throughput as the coupled setup, demonstrating that the decoupled architecture itself does not introduce any overhead. These results showcase Wirehead's ability to efficiently decouple generation and training, even enhancing generation speed in a distributed environment.

The overall sample throughput is summarized in Table II.

### D. Linear Scaling of Sample Generation

Our results demonstrate near-linear scaling of sample generation rate with respect to the number of worker nodes. Doubling the workers from 1 to 2 yields a 1.99x throughput increase, while quadrupling to 4 results in a 3.91x increase. Further scaling to 8 and 16 workers achieves 7.75x and 15.70x speedups, respectively (Table III).

This linear scaling behavior showcases Wirehead's excellent scalability, enabling efficient distribution of workload across multiple nodes with minimal overhead. The ability to scale

generation throughput linearly is crucial for handling large-scale machine learning tasks requiring vast amounts of synthetic data. Wirehead's architecture allows users to easily accommodate growing data requirements and directly translates to faster experiment turnover.

### E. Training performance

To ensure a fair comparison across configurations, we conducted all training experiments on identical hardware: a single NVIDIA A100 GPU node with 16 CPU cores and 200GB of RAM. Each experiment ran for a duration of 5 hours, providing a consistent time frame for performance evaluation. (Figure 3)

While the training setup remained constant, the data generation configurations varied. For both the coupled and local Wirehead setups, data generation occurred on the same A100 node as training. In contrast, the distributed Wirehead configuration leveraged 20 separate nodes for data generation, significantly increasing the potential throughput of synthetic data production.

All evaluation scores were obtained using real unseen Human Connectome Project (HCP) data. Utilizing these configurations, we obtained the following evaluation results:

- Coupled: 0.39 Evaluation Dice Score

- Local Wirehead: 0.28 Evaluation Dice Score
- Distributed Wirehead: 0.80 Evaluation Dice score

The traditional coupled approach demonstrated extremely slow growth in DICE scores on the HCP evaluation data. By the end of our experiment, the model had not yet reached saturation, indicating a significantly slower learning process compared to the Wirehead configurations. This slow progress underscores the inefficiencies inherent in coupling data generation with model training, where the model often waits for new data to be generated before it can continue learning.

The local Wirehead configuration showed a rapid increase in DICE scores for the training set (generated data). However, when evaluated on the real HCP dataset, the scores plateaued quickly at around 0.28. This discrepancy between training and evaluation performance is noteworthy and suggests potential overfitting to the generator itself.

We hypothesize that this overfitting is due to the significant imbalance between read throughput and generation throughput in this setup. With read operations occurring multiple times faster than generation, each generated sample is seen approximately 3.5 times on average during training. This repetition likely leads to the model memorizing specific patterns in the data generation process rather than learning generalizable features, resulting in poor performance on real-world HCP data.

The distributed Wirehead configuration demonstrated the most promising results, showing rapid improvements in both HCP evaluation and training DICE scores. Notably, it surpassed the HCP evaluation score achieved by the coupled setup at the 5-hour mark in just 40 minutes, a significant 7x speedup. The distributed configuration ultimately peaked at a DICE score of 0.8 on the HCP data, substantially outperforming both the coupled and local configurations.

### F. Key Findings

1) GPU Utilization: Wirehead significantly improved GPU utilization compared to the coupled implementation, with the distributed setup showing the highest efficiency.
2) Read and Generation Speed: Both local and distributed Wirehead setups showed substantial improvements in samples read and generated per second compared to the coupled implementation.
3) Scalability: The near-linear scaling of sample generation with worker count demonstrates Wirehead's excellent scalability in distributed environments.
4) Decoupling Benefits: The decoupled architecture allowed for independent optimization of generation and training processes, leading to overall system performance improvements.
5) Scaling Impact on Training: Distributed Wirehead's ability to scale data generation across multiple nodes led to a significant speedup in model performance growth and a higher final DICE score (0.8) on real HCP data.

These benchmarks clearly demonstrate the advantages of Wirehead's architecture, particularly in distributed environments. The system's ability to scale linearly with additional resources makes it well-suited for large-scale machine learning tasks requiring substantial synthetic data generation.

## VII. Discussion

### A. Swap Cap and Data Freshness

The swap cap in Wirehead plays a crucial role in managing data flow and system performance. While one might initially assume that a larger swap cap would provide a bigger cache and thus better performance, our analysis reveals a more nuanced reality:

1) Cache Size vs. Swap Frequency: A larger swap cap does indeed provide a bigger cache. However, it also means less frequent swaps, by exactly the same factor. This trade-off effectively cancels out the potential benefits of a larger cache in many scenarios.
2) Effective Read Rate: Interestingly, the rate of repeated reads per sample remains relatively constant regardless of the swap cap configuration. This is because the slower swap rate with a larger cap counterbalances the increased cache size.
3) Memory Considerations: The memory required to store all variables scales linearly with the swap cap. The formula is approximately: Memory = swap_cap * sample_size * 2 (for read and write caches)
   In our tests, we successfully scaled up to 10,000 samples (uint8, 256x256x256 tensors, in data-label pairs), consuming about 620 GB of memory without performance issues.
4) Optimal Configuration: The ideal swap cap depends on the specific use case, balancing memory availability, data freshness requirements, and the relative speeds of generation and training. In general, we find that larger swap caps lead to slightly better hardware utilization due to having fewer swaps from the write cache to the read cache. Further experiments are required to assess the impact of different configurations on model performance.

### B. Generation Throughput and Training Balance

The decoupled nature of generation and training in Wirehead leads to two potential scenarios, each with its own implications:

1) Generation is Faster than Training:
   - Pros: Increased data variance, potentially harder to overfit.
   - Cons: Possible waste of compute resources for generation.
   - Implication: Training effectively samples from a subset of generated data.
2) Training is Faster than Generation:
   - Issue: Samples may be seen multiple times during training.
   - Risk: Potential overfitting, especially for certain training tasks.

- Example: In our local Wirehead experiment, training was 4 times faster than generation, resulting in each sample being seen an average of 4 times. This led to a lower evaluation DICE score (see Appendix for details).

These observations highlight the importance of carefully balancing generation and training speeds. While Wirehead's architecture allows for independent scaling, optimal performance requires thoughtful configuration based on the specific requirements of the task at hand.

Potential Mitigations:

1) Dynamic swap cap adjustment based on observed generation and training speeds.
2) Implementing a "freshness" policy that prioritizes newer samples in the training process.
3) For cases where generation is faster, consider implementing a sampling strategy that ensures broader coverage of the generated data.
4) When training is faster, explore techniques to mitigate overfitting, such as increased regularization or dynamic learning rate adjustment.

These findings underscore the complexity of managing large-scale synthetic data generation and training processes. While Wirehead provides a powerful and flexible framework, achieving optimal performance requires careful consideration of these interdependencies and potential trade-offs.

## VIII. Software Availability and Ease of Use

To facilitate the adoption and reproducibility of our work, we have made Wirehead available both as open-source software under MIT license, and as a pip-installable package. This approach significantly streamlines the setup process for researchers and practitioners interested in utilizing our caching system for scaling synthetic data generators.

The installation process is straightforward and can be completed with a single command:

```
pip install wirehead
```

This installation method ensures that all necessary dependencies are automatically managed, reducing the potential for configuration errors and version incompatibilities.

Furthermore, we have prioritized user convenience by providing comprehensive documentation and usage instructions in our GitHub repository[1]. These resources offer step-by-step guidance on configuring and integrating Wirehead into existing data generation pipelines.

## IX. Rapid Model Development

Wirehead's rapid experimentation capabilities have opened up new possibilities for model development and deployment. A prime example is Brainchop[2] [16], a brain segmentation platform that allows models to run natively in the browser. With Wirehead, we can rapidly iterate through model architectures,

[1] Available at: https://github.com/neuroneural/wirehead/tree/paper
[2] Available at: https://brainchop.org/

such as the MeshNet [17], model sizes, and alternative use cases, such as tissue extraction.

## X. Conclusions

This paper presents Wirehead, a novel system for horizontally scaling synthetic data generation and training in machine learning. Wirehead's key innovations include:

- A decoupled architecture enabling horizontal scaling with near-linear performance gains relative to worker count.
- An efficient data management system utilizing MongoDB, capable of handling terabytes of data with minimal I/O or processing overhead.
- A robust, modular design allowing for flexible pipeline configurations and fault-tolerant operation.

These features collectively accelerate the experimentation cycle, enabling faster iteration in data-intensive machine learning tasks. While Wirehead demonstrates significant advantages in scalability and efficiency, it also reveals opportunities for future research:

- Developing dynamic resource allocation strategies to optimize the balance between data generation and training.
- Exploring Wirehead's applicability in other domains with similar challenges in data characteristics, notably large data size and long processing speeds. Notable examples of interest are training on synthetic video generation pipelines, and distilling large language models weights.

By enabling efficient and scalable deployment of synthetic data generation, Wirehead has the potential to accelerate progress in machine learning research and applications, particularly in fields constrained by data availability. However, realizing this potential will require ongoing refinement of the system and application across diverse domains.

## Acknowledgment

Special thanks go to Armina Fani for her uncanny ability to uncover bugs in our software, ultimately making our implementations more robust, and to Thu Le for providing additional labeled data for out-of-domain experiments, which highlighted the challenges of relying on synthetic data for deep learning. We are also grateful to Mateo Sanabria for his constant encouragement and willingness to engage in rubber duck debugging sessions. We would also like to acknowledge Alex Fedorov for his expertise in organizing deep learning projects.

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
