# OpenReview forum: "Scaling Synthetic Brain Data Generation"
_IEEE.org/EMBS/BHI/2024/Conference — IEEE BHI'24_

### Official Review · Reviewer_bbHV · 2024-08-14
**Enhancing GPU Efficiency: A Decoupled Pipeline for On-the-Fly Data Generation in Neuroimaging**

**Overall Rating:** 8
**Confidence:** 4

**Other Quality Metrics:**

(a) Clarity of writing:good
(b) Clinical Significance:good
(c) Methodological Novelty: excellent
(d) Experiments and Results: great

**Questions For The Authors:**

Literature review: Beyond the proposed pipeline and the traditional coupled method, are there any other approaches for on-the-fly data generation and training?

**Strengths:**

Strong engineering approach to enhancing on-the-fly data generation and training.
Well-articulated background introduction.
Clear and logical writing structure.
Comprehensive and well-reasoned result analysis.
Practical solutions with promising potential and thoughtful future work considerations

**Summary Of The Paper:**

This paper proposes an on-the-fly synthetic data generation and training pipeline that significantly improves GPU computing efficiency, accelerating deep learning model training in neuroimaging. By utilizing MongoDB for cache swapping and decoupling data generation from training, the pipeline enables flexible and independent operations for both processes. Additionally, it highlights opportunities for future research in optimizing the balance between data generation and training, as well as resource allocation in tasks requiring large-scale, on-the-fly generation of synthetic data for training.

**Weaknesses:**

Incomplete pipeline description: The authors did not address scenarios where the training process finishes while data generation is still ongoing until they analyzed the low Dice Score for Local Brainpipe. This omission led to some confusion during the initial reading. It would be clearer to mention this scenario when describing the pipeline.

---

### Official Review · Reviewer_LKKd · 2024-08-14
**Simultaneous data generation and training improve efficiency but lack statistical validation and broader comparisons. Needs to address some issues mentioned in this review**

**Overall Rating:** 8
**Confidence:** 5

**Other Quality Metrics:**

(a) Clarity of writing; great
(b) Clinical Significance; excellent
(c) Methodological Novelty; great
(d) Experiments and Results: great

**Questions For The Authors:**

1. The Dice score results in Fig3 appear promising. Would be better to add some other ML task such as classification, regression and report respective metric
2. VII. Discussion B: Understandable that "Training is Faster than Generation" leads to poor generalization. Is there a sweet spot? How is it changing with the data type/domain
3. Fig 2 has poor quality, font size and color contrast.

**Strengths:**

The paper introduces a novel approach to developing machine learning models by simultaneously generating data and training the models. This method is particularly promising due to its computational efficiency, as it significantly reduces training time compared to traditional methods that first generate the entire training dataset and then use it for training. The approach is backed by experimental results that demonstrate several benefits, including enhanced GPU utilization, reduced data storage requirements, faster processing times, and higher dice scores in ML tasks. These aspects suggest that the proposed method could offer substantial improvements in training efficiency while allowing more time for tuning hyperparameters.

**Summary Of The Paper:**

This paper presents an innovative approach to developing machine learning models where the data is generated and used for training simultaneously. The traditional coupled training and data generation processinvolves first generating the entire training dataset and then using it for training. This method can consume significant storage and computational resources, leading to extended training times. However, the authors' proposed approach has proven to be computationally efficient, significantly reducing training time. Overall, the paper offers a compelling perspective, supported by experimental results that demonstrate enhanced GPU utilization, reduced data storage requirements, faster processing times, and higher dice scores in their ML tasks.

**Weaknesses:**

While the paper presents a promising approach, it lacks a thorough experiments and discussion on statistical implications of this approach such as generalization of the ML models, mods in the subsets of generated data, and any potential challenges in reproducing the results. How the data generation is being controlled to ensure the iid nature of training sample? A wider comparative study based on utilization of ML models for various tasks such as classification, regression is needed to ascertain their claims about the efficacy of their approach. Controlling the randomness i.e. fixing the seeds may be valuable for reproducibility.

---

### Official Review · Reviewer_TT83 · 2024-08-15
**The paper presents good results.**

**Overall Rating:** 7
**Confidence:** 2

**Other Quality Metrics:**

(a) Clarity of writing: great
(b) Clinical Significance: good
(c) Methodological Novelty: good
(d) Experiments and Results: good

**Questions For The Authors:**

How to verify the Brainpipe's ability in other areas?

**Strengths:**

The authors proposed the  Brainpipe, an innovative in-memory data pipeline designed to facilitate model-agnostic, scalable, and real-time synthetic data generation for deep learning.

**Summary Of The Paper:**

The author presents the Brainpipe, a scalable in-memory data pipeline that dramatically boosts the efficiency of real-time synthetic data generation for deep learning applications in neuroimaging.

**Weaknesses:**

The related work needs to be compared.

---

### Decision · Program_Chairs · 2024-09-23

Accept